# A systematic review of methodological approaches for evaluating real-world effectiveness of COVID-19 vaccines: Advising resource-constrained settings

Yot Teerawattananon[1,2], Thunyarat Anothaisintawee[3], Chatkamol Pheerapanyawaranun[1], Siobhan Botwright[1]*, Katika Akksilp[1], Natchalaikorn Sirichumroonwit[1], Nuttakarn Budtarad[1], Wanrudee Isaranuwatchai[1]

1 Health Intervention and Technology Assessment Program, Ministry of Public Health, Nonthaburi, Thailand,
2 Saw Swee Hock School of Public Health, National University of Singapore, Singapore, Singapore,
3 Department of Family Medicine, Faculty of Medicine, Ramathibodi Hospital, Mahidol University, Bangkok, Thailand

* siobhan.b@hitap.net

**Data Availability Statement:** All articles included in the review are listed in the manuscript and

## Abstract

Real-world effectiveness studies are important for monitoring performance of COVID-19 vaccination programmes and informing COVID-19 prevention and control policies. We aimed to synthesise methodological approaches used in COVID-19 vaccine effectiveness studies, in order to evaluate which approaches are most appropriate to implement in low- and middle-income countries (LMICs). For this rapid systematic review, we searched PubMed and Scopus for articles published from inception to July 7, 2021, without language restrictions. We included any type of peer-reviewed observational study measuring COVID-19 vaccine effectiveness, for any population. We excluded randomised control trials and modelling studies. All data used in the analysis were extracted from included papers. We used a standardised data extraction form, modified from STrengthening the Reporting of OBservational studies in Epidemiology (STROBE). Study quality was assessed using the REal Life EVidence AssessmeNt Tool (RELEVANT) tool. This study is registered with PROSPERO, CRD42021264658. Our search identified 3,327 studies, of which 42 were eligible for analysis. Most studies (97.5%) were conducted in high-income countries and the majority assessed mRNA vaccines (78% mRNA only, 17% mRNA and viral vector, 2.5% viral vector, 2.5% inactivated vaccine). Thirty-five of the studies (83%) used a cohort study design. Across studies, short follow-up time and limited assessment and mitigation of potential confounders, including previous SARS-CoV-2 infection and healthcare seeking behaviour, were major limitations. This review summarises methodological approaches for evaluating real-world effectiveness of COVID-19 vaccines and highlights the lack of such studies in LMICs, as well as the importance of context-specific vaccine effectiveness data. Further research in LMICs will refine guidance for conducting real-world COVID-19 vaccine effectiveness studies in resource-constrained settings.

excluded articles are available in supplementary file. The data extraction form has been uploaded.

**Funding:** This study was funded by the Health Systems Research Institute (https://hsri.or.th/researcher), grant number 64134002RM011L0. The funder of the study had no role in study design, data collection, data analysis, data interpretation, or writing of the report.

**Competing interests:** This study was funded by the Health Systems Research Institute (https://hsri.or.th/researcher), grant number 64134002RM011L0. The authors declare that no other competing interests exist.

## Introduction

The COVID-19 pandemic has placed a significant toll on health systems and economies. With the development and roll-out of COVID-19 vaccines, policymakers in low- and middle-income countries (LMICs) now have an additional tool to control the pandemic, with the potential to ease lockdowns and other non-pharmaceutical interventions. Yet there is increasing evidence to suggest that vaccines are not a magic bullet, and policymakers will have to identify how to best use vaccines as part of a comprehensive set of interventions [1]. In the immediate term, vaccination programme constraints, both in terms of vaccine supply as well as the capacity of health programmes to deliver vaccine at an unprecedented scale, mean that policymakers must identify how best to target vaccines for greatest impact. In the longer-term, financial sustainability is likely to become an ever more pressing issue. Policymakers have been able to allocate emergency funding to finance COVID-19 prevention and control measures, and many financial institutions have unlocked access to grants and concessional loans to tackle the pandemic [2]. However, as more data become available on vaccine duration of protection, protection against transmission, and protection against COVID-19 variants, policymakers will have to decide which vaccination strategies are sustainable and most appropriate to implement in their context [3]. Already there are stark differences in COVID-19 vaccination coverage targets between countries, ranging from those aiming to vaccinate 30% of the population to those aiming for full population coverage [4].

To inform evidence-based policies on the rational use of COVID-19 vaccines, LMICs require real-world data on the effectiveness of vaccines in their context. Efficacy data from clinical trials are important for regulatory authorities to identify if a vaccine works and if it is safe. However, there are a number of limitations in using efficacy data for policy. Firstly, clinical trials use strict inclusion and exclusion criteria, which are not necessarily representative of all eligible populations for vaccination [5–7]. For COVID-19, a number of vaccines have been recommended for use with limited data on effectiveness in the elderly, pregnant women, and populations with comorbidities, despite these being priority target groups in many countries [8–11]. Second, the setting of clinical trials may not reflect local epidemiology. COVID-19 vaccine clinical trials have been conducted in settings with different circulating strains, diverse underlying population health, varying transmission dynamics and non-pharmaceutical interventions (NPIs), and measuring different outcomes [12]. Finally, due to their nature, efficacy studies are unable to address programmatic issues around health service utilization or off-label use [5]. For COVID-19 vaccines, this includes issues such as timely receipt of the second dose, modified vaccine schedules to address supply shortages or to align timing across vaccine products, vaccine acceptance and hesitancy (especially among specific population groups), interchangeability for mixed product schedules, cold chain excursions and other logistics issues, among others [13].

Real-world effectiveness studies are important for informing policy decisions, as an estimate of the context-specific performance of vaccines [13–15]. The results from real-world effectiveness studies not only monitor impact, but also give country-specific inputs for modelling future strategies for vaccination and relaxation of NPIs, as well as justifying budget allocation into, or away from, the COVID-19 vaccination programme. Due to the nature of real-world effectiveness studies, they can be subject to selection bias, confounding factors, and missing data, therefore requiring careful study design [5, 16, 17]. Important considerations for observational studies include sample size; methods to minimise selection bias; accurate measurement of exposures and outcomes; planning for, managing, and reporting on potential confounders and missing data; and planning appropriate analysis [16, 17].

The World Health Organization (WHO) has published an interim guidance for conducting vaccine effectiveness studies in LMICs, and is maintaining a landscape of observational study designs for COVID-19 vaccination effectiveness [13, 18]. Whilst many studies have synthesised COVID-19 vaccine effectiveness estimates from observational studies [19–24], to our knowledge, there is no systematic review of published real-world effectiveness study designs for COVID-19 vaccination, to support LMICs to understand which study designs are most feasible to implement in their settings, and the advantages and drawbacks of different approaches. This review was commissioned by the Thai government to summarise methodological approaches being used to study real-world COVID-19 vaccine effectiveness, to assess the quality of published literature, and to consider which best-practice approaches are most suitable for implementation in Thailand and other LMICs.

## Methods

### Search strategy and selection criteria

We conducted a systematic review of the literature to identify peer-reviewed research studies on COVID-19 vaccine effectiveness, in order to analyse the study design and methods for applicability to LMICs. We chose a rapid review methodology as a streamlined approach to quickly inform policymakers and researchers in Thailand and other LMICs that are in the process of developing vaccine effectiveness studies. Since the objective of the review was to analyse methodological approaches, we did not conduct meta-analysis to summarise the results.

We included research studies published in academic journals in any language, which reported on the effectiveness of COVID-19 vaccination in real-world settings. We therefore included any type of observational study, including cohort studies (prospective and retrospective), case control studies, test-negative design case-control studies, and screening studies, but excluded randomised control trials (RCTs) and modelling studies. We also excluded regression discontinuity design as it is currently recommended for vaccine effectiveness studies in diseases with low incidence, or for which there is a long time lag until the outcome [25]. Primary research articles were eligible, as were letters to the editor, correspondence, reports, or rapid communications, provided that the methods were adequately described for data extraction and quality assessment of study design. Due to our focus on methodological approaches, we only included peer-reviewed literature, as quality assurance for study design and reporting. We did not exclude studies based on population of interest, but restricted inclusion to studies measuring the following outcomes: asymptomatic SARS-CoV-2 infection, symptomatic SARS-CoV-2 infection, severe SARS-CoV-2 infection (as measured by hospital admission, ICU admission, or clinical diagnosis), or death from SARS-CoV-2 infection.

We executed a search strategy (S1 Appendix) of articles published from inception to July 7, 2021, in the MEDLINE (via PubMed) and Scopus databases. Search terms were constructed according to intervention of interest (COVID-19 vaccine) and study design (e.g. cohort study, post-marketing study, effectiveness analysis). Searching the reference lists of the included studies and consultation with experts identified additional relevant studies. In the first stage, titles and abstracts were screened independently by two reviewers, each from one of two separate teams. Any disagreement was resolved by one of two reviewers (YT or TA). In the second stage, full text was reviewed for inclusion/exclusion by a single reviewer.

### Data analysis

All authors extracted data using a structured form modified from STrengthening the Reporting of OBservational studies in Epidemiology (STROBE), the reporting standard for observational studies [26]. Data were abstracted on study characteristics (objectives, type of study

design, country, study duration, funding source); study sample (population, sample size, presence of variants of concern); intervention (partial or full vaccination, vaccine product received); study outcomes; data collection and measurement methods (including utilisation of existing database); data analysis methods (subgroup analysis, statistical model, sensitivity analysis, management of missing data and potential confounders); results (by outcome of interest); study limitations; and ethical approval and/or consent requirements. Type of study design was classified by the authors based on definitions from the WHO interim guidance on evaluation of COVID-19 vaccine effectiveness [13]. For the results, vaccine effectiveness (%) by outcome was recorded. For studies reporting incidence rate ratio (IRR), the formula (1-IRR)*100 was used to calculate vaccine effectiveness. The quality of studies was assessed by two independent reviewers using the REal Life EVidence AssessmeNt Tool (RELEVANT) tool [27]. Each primary and secondary sub-item was scored as 1 (yes) if performed or reported in the study, otherwise a score of 0 (no) was assigned. Two reviewers (YT and TA) resolved any discrepancy in scoring. Qualitative analysis of results from using the RELEVANT tool identified areas of limited evidence and highlighted opportunities to strengthen COVID-19 vaccine effectiveness study methodology.

Figures were produced using R, version 4.1.0 (Camp Pontanezen). The review protocol is registered at PROSPERO, CRD42021264658.

## Results

We identified 5,933 articles through the database search. No additional articles were identified from searching reference lists. After removal of duplicates (2,606) and exclusion of studies based on screening the abstract (3,249) or the full text (42), 36 studies were identified. We included an additional 6 studies identified during expert consultation, resulting in 42 papers for inclusion (Fig 1). Of the 42 studies excluded during full text screening, 31 reported on an excluded outcome (not effectiveness) and 11 were an excluded study type (randomised control trial or modelling study). All studies were in English, except one study in Spanish.

All 42 studies identified were published in 2021 and all but one study [28] were conducted in high-income countries (HICs) (Table 1). No studies were identified from Africa and only one from Asia [28]. Presence of circulating variants were reported in 12 (29%) studies [11, 29–39]. Most studies assessed effectiveness of mRNA vaccines (33 studies), followed by an mRNA and a viral vector vaccine (7 studies), and 1 study each for viral vector and inactivated vaccine. Ethical approval was required in 27 studies (64%), with 13 studies (31%) not reporting on ethical approval. Many studies (18, 43%) did not report on funding source; for the other studies, 11 (26%) were publicly funded, 2 (5%) funded through public and private funds, 3 (7%) through not-for-profit private funding, and 8 (19%) did not receive funding.

Table 2 summarises study characteristics. Most studies (32 of 42, 76%) reported on vaccine effectiveness against either COVID-19 infection, hospitalisation, or death, whereas 3 studies reported 2 outcomes (hospitalisation and infection [37, 66], hospitalisation and death [51]) and 7 studies reported on all 3 outcomes [31, 33, 35, 42, 54, 58, 59]. Of the 37 studies measuring vaccine effectiveness against infection, 31 are cohort studies, 4 test-negative design case control studies, and 2 screening method (Fig 2). The most common study type is retrospective cohort study, (22 studies), often employing immunisation registries and medical databases. Only five studies considered asymptomatic infection among patients under investigation, frontline workers and randomly selected individuals in the community [11, 37, 39, 61, 62]. Most cohort studies were conducted among healthcare workers undergoing routine RT-PCR testing as part of the hospital surveillance system. Sample size ranged from 189 to 10,187,720 (mean 443,697; median 6,904). For vaccine effectiveness against hospitalisation and/or death,

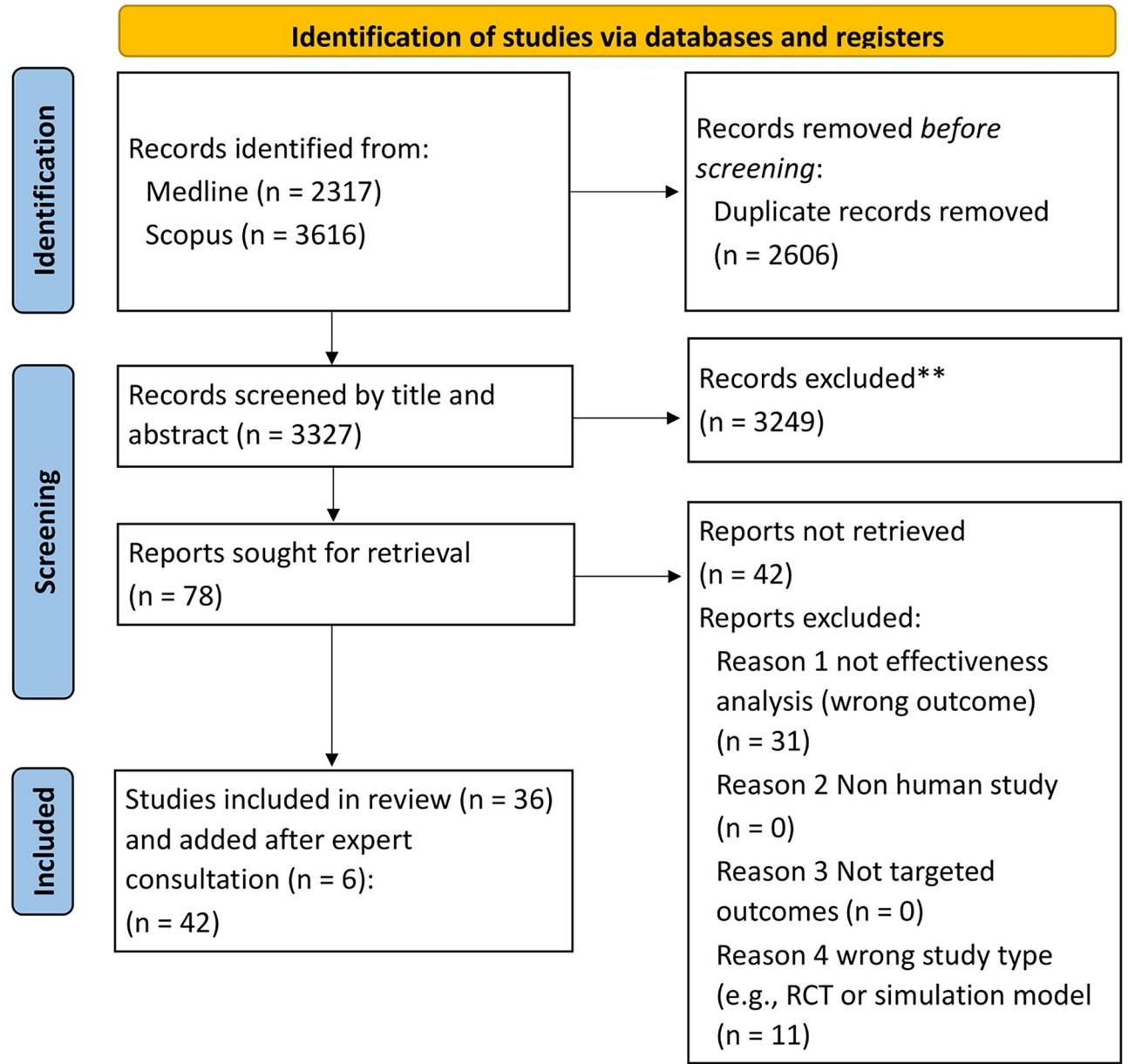

**Fig 1. Study profile.**

we identified 12 cohort and 2 test negative design case control studies. Contrary to infection studies, none had healthcare workers as the population. All studies in the general population used national level surveillance data. Sample size ranged from 189 to 10,187,720 (mean 1,890,171; median 338,145). The test negative designs had small sample sizes compared to cohort studies.

Table 3 summarises methodology employed across included studies. Most studies assessed vaccination status by registry (31), with 2 studies using self- report [9, 41], 3 using a mixture of registry and self-report [35, 44, 60], and 6 studies not reporting on methods to ascertain vaccination status [28, 32, 47, 61, 64, 65]. For confirmation of COVID-19 infection, 39 studies confirmed diagnosis with reverse transcription polymerase chain reaction (RT-PCR); 2 studies

**Table 1.  General characteristics of articles on real-world effectiveness of COVID-19 vaccines.**

| Characteristics | N (%) |
|---|---|
| **Publication year** | |
| 2021 | 42 (100%) |
| **Publication type** | |
| Correspondence | 4 (9%) |
| Letter | 3 (7%) |
| Original (primary) research | 29 (70%) |
| Rapid communication | 4 (9%) |
| Report | 2 (5%) |
| **Country** | |
| Chile | 1 (2.5%) |
| Qatar | 1 (2.5%) |
| India | 1 (2.5%) |
| Ireland | 1 (2.5%) |
| Israel | 9 (21%) |
| Italy | 3 (7%) |
| Scotland | 1 (2.5%) |
| Spain | 4 (9%) |
| United Kingdom | 7 (17%) |
| United States | 13 (31%) |
| Multinational | 1 (2.5%) |
| **Vaccine types** | |
| mRNA (BNT162b2) | 25 (59%) |
| mRNA (mRNA-1273) | 2 (5%) |
| mRNA (BNT162b2 and mRNA-1273) | 6 (14%) |
| mRNA and viral vector (BNT162b2 and ChAdOx1-S) | 5 (12%) |
| mRNA and viral vector (BNT162b2, mRNA-1273 and ChAdOx1-S) | 2 (5%) |
| Viral vector (ChAdOx1-S and BBV152) | 1 (2.5%) |
| Inactivated SARS-CoV-2 (CoronaVac) | 1 (2.5%) |
| **Variants** | |
| Mentioned | 12 (29%) |
| B.1.1.7 (alpha) | 8 |
| B.1.1.7 and B.1.351 | 2 |
| B.1.1.7 and B.1.525 | 1 |
| R.1 lineage | 1 |
| Not mentioned | 30 (71%) |
| **Ethical approval** | |
| Yes | 27 (64%) |
| Exempted | 2 (5%) |
| Not stated | 13 (31%) |
| **Informed consent** | |
| Yes | 2 (5%) |
| Exempted | 7 (17%) |
| Full ethical review was not necessary | 8 (19%) |
| Not stated | 25 (59%) |
| **Study design** | |
| Test-negative design case control study | 5 (12%) |
| Prospective cohort study | 7 (17%) |

(*Continued*)

**Table 1.** (Continued)

| Characteristics | N (%) |
|---|---|
| Retrospective cohort study | 28 (66%) |
| Screening methods | 2 (5%) |
| **Outcomes (a study can have more than one outcome)** | |
| Infections | 37 |
| Hospitalizations | 10 |
| Mortality | 9 |
| **Financial source** | |
| Public | 11 (26%) |
| Public and Private | 2 (5%) |
| Private not for profit | 3 (7%) |
| None | 8 (19%) |
| Not reported | 18 (43%) |

used RT-PCR as the main method of confirming diagnosis, but either allowed rapid antigen test for symptomatic cases [37] or if RT-PCR was not available [54]; and 1 study did not mention method of confirmation of COVID-19 [28]. Of the studies reporting methods to reduce misclassification error, most restricted analysis to samples collected within a certain number of days from symptom onset, ranging up to 7 days before symptom onset and 7–14 days after symptom onset [9, 10, 33, 35, 37, 46, 50]. Other studies reported reducing misclassification error by restricting analysis to symptomatic cases [9, 42, 46], censoring the date of unreliable vaccination dates [11], and conducting sensitivity analysis removing days for possible misclassification [60]. Although not reported as a method to reduce misclassification error, an additional 12 studies only included symptomatic cases [10, 32, 34, 35, 39, 47, 50, 53, 54, 58]. There was considerable difference across studies in terms of when outcomes were assessed in vaccinated individuals: 10 studies only included outcomes more than 14 days after vaccination [31–33, 37, 39, 44, 49, 54, 57, 60]; 10 studies more than 7 days after vaccination [10, 34, 35, 42, 46, 47, 53, 55, 63, 66]; 9 studies included outcomes more than 14 days after vaccination for one of the two vaccine doses, and more than 7 days after the other vaccine dose [9, 11, 30, 38, 45, 50, 51, 56, 62]; 2 studies included outcomes either 14 days or 7 days after vaccination depending on vaccine type [48, 59]; 7 studies included outcomes any time after vaccination, but stratified outcomes by number of days after vaccination [36, 40, 41, 43, 52, 58, 64]; 2 studies included outcomes any time after vaccination [29, 65]; and 2 studies did not report on time between vaccination and outcome inclusion [28, 61]. 3 studies conducted sensitivity or sub-group analysis by days after vaccination [46, 49, 65].

For the quality assessment using RELEVANT, 9 of the 42 studies (of which all were cohort studies) met less than half of the criteria [28, 34, 43, 47, 48, 53, 61, 63, 64]. Only 10 of the 43 studies reported registration or publication of the study protocol and 17 reported on potential conflicts of interest (Fig 3). Regarding study methods, there were a number of limitations across studies. Firstly, due to the short time since vaccine roll-out, follow-up time for all studies was very short (mean 6.3 weeks for studies with infection outcomes, 9.7 weeks for hospitalisation or death outcomes). Secondly, only 10 studies reported calculating a sample size a priori (Fig 3). Although studies with large national datasets do not need to calculate a minimum sample size, 3 out of 4 (75%) of the test negative case control designs with fewer than 5,000 participants did not report calculating a minimum sample size [9, 10, 41], and this was also the case for 6 out of 10 of the cohort studies with fewer than 5,000 participants [30, 47, 55, 57, 64, 65]. Thirdly, most studies did not clearly delineate inclusion/exclusion of study participants as a

**Table 2. Characteristics of COVID-19 vaccine real-world effectiveness studies meeting inclusion criteria.**

| | Country | Funding source | Population | Sample size | Study design* | Study time frame | Database(s) | Type(s) of vaccine | Outcome |
|---|---|---|---|---|---|---|---|---|---|
| Lopez-Bernal [33] | U.K. | None | Elderly people aged ≥70 years old | 265,745 | Test negative case-control design | October 26, 2020—February 21, 2021 | National Immunisation Management System and hospital admission data | BNT162b2, ChAdOx1-S | SAR-CoV2 infection, hospital admissions, deaths |
| Vasileiou [40] | U.K. | UK Research and Innovation (Medical Research Council), Research and Innovation Industrial Strategy Challenge Fund, Health Data Research UK | General population | 5.4 million | Prospective cohort study | December 8, 2020—February 22, 2021 | Early Pandemic Evaluation and Enhanced Surveillance of COVID-19—EAVE II—database, Scottish Morbidity Record 01 database, and Rapid Preliminary Inpatient Data. | BNT162b2, ChAdOx1-S | Hospital admissions due to SARS-CoV-2 infection |
| Tenforde [41] | USA | Not stated | Adults with COVID-19–like illness admitted to 24 hospitals in 14 states. Patients were eligible if they were ≥65 years on the date of hospital admission, received clinical testing for SARS-CoV-2 by RT-PCR or antigen test within 10 days of illness onset, and had onset of symptoms 0–14 days before admission. | 417 | Observational study | January 1–March 26, 2021 | Not stated | BNT162b2 | SAR-CoV2 infection and hospital admissions |
| Haas [42] | Israel | Israel MoH and Pfizer | >16 years old residents of Israel | Isreali population in 1 of 4 nationwide medical insurance programmes | Observational study | January 24—April 3, 2021 | Nationwide Surveillance Data | BNT162b2 | SAR-CoV2 infection, hospital admissions, deaths |
| Sansone [34] | Italy | Not stated | Healthcare workers in Brescia | 6,904 | Observational study | January 25, 2021—April 13, 2021 | No database used | BNT162b2 | SAR-CoV2 infection |
| Keehner [43] | USA | Not stated | Healthcare workers in University of California, San Diego (UCSD) and University of California, Los Angeles (UCLA) | 36,659 | Observational study | December 16, 2020 – February 9, 2021 | Electronic employee health record system at UCSD and UCLA | BNT162b2, mRNA 1273 | SAR-CoV2 infection |

*(Continued)*

**Table 2.** (*Continued*)

| | Country | Funding source | Population | Sample size | Study design* | Study time frame | Database(s) | Type(s) of vaccine | Outcome |
|---|---|---|---|---|---|---|---|---|---|
| Thompson [44] | USA | Not stated | Healthcare workers, first responders, and frontline workers | 3,950 | Observational study | December 14–18, 2020—March 13, 2021. | No database used | BNT162b2, mRNA 1273 | SAR-CoV2 infection |
| Fabiani [45] | Italy | Not stated | Frontline healthcare workers | 6,423 | Retrospective cohort study | December 27, 2020—March 24, 2021 | Local COVID-19 surveillance database | BNT162b2 | SAR-CoV2 infection |
| Cavanaugh [35] | USA | Not stated | Residents and healthcare workers | 189 | Retrospective cohort study | January 10—March 1, 2021 | Immunization registry review; facility interviews; medical records reviews | BNT162b2 | SAR-CoV2 infection, symptomatic COVID-19 cases, hospital admissions, deaths |
| Hall [11] | U.K. | Public Health England, UK Department of Health and Social Care, and the National Institute for Health Research | Healthcare workers and staff ≥18 years old | 23,324 | Prospective cohort study | Dec 7, 2020—Feb 5, 2021 | Participants enrolling to the National Immunization Management System | BNT162b2 | SAR-CoV2 infection |
| Benenson [36] | Israel | Not stated | Healthcare workers | 6,680 | Descriptive cohort study | 8 weeks after Dec 20, 2020 | Not stated | BNT162b2 | SAR-CoV2 infection |
| Martínez-Baz [37] | Spain | The Horizon 2020 program of the European Commission and the Carlos III Institute of Health with the European Regional Development Fund | Individuals aged ≥18 years covered by the Navarre Health Service with close contacts of laboratory-confirmed COVID-19 cases | 20,961 | Prospective cohort study | January to April 2021 | Not stated | BNT162b2, ChAdOx1-S | SAR-CoV2 infection |
| Chodick [46] | Israel | Not stated | All Maccabi Healthcare Services (MHS) members aged 16 years or older who were vaccinated during a mass immunization program | 503,875 | Comparative effectiveness study | December 19, 2020—January 15, 2021 | Maccabi Healthcare Services | BNT162b2 | SAR-CoV2 infection |
| Jameson [47] | USA | None | All healthcare workers in a hospital | 4,318 | Screening | December 17, 2020—March 24, 2021 | Not stated | BNT162b2 | SAR-CoV2 infection |

(*Continued*)

**Table 2.** (Continued)

| | Country | Funding source | Population | Sample size | Study design* | Study time frame | Database(s) | Type(s) of vaccine | Outcome |
|---|---|---|---|---|---|---|---|---|---|
| Pilishvili [9] | USA | Not stated | Routine employee testing performed based on site-specific occupational health practices. | 1,843 | Test negative case-control study | January–March 2021 | Not stated | BNT162b2, mRNA 1273 | SAR-CoV2 infection |
| Daniel [48] | USA | Texas Department of State Health Services | University employees | 23,234 | Descriptive data report | December 15, 2020—February 28, 2021 | University of Texas Southwestern Medical Center (UTSW) | BNT162b2, mRNA 1273 | Decrease in the number of employees who are either in isolation or quarantine and reduction in the incidence of infections |
| Angel [38] | Israel | Not stated | Healthcare workers | 6,710 | Retrospective cohort study | December 20, 2020—February 25, 2021 | Hospital data | BNT162b2 | SAR-CoV2 infection |
| Amit [49] | Israel | Not stated | Healthcare workers | 9,109 | Retrospective cohort study | December 19, 2020—January 24, 2021 | Not stated | BNT162b2 | SAR-CoV2 infection |
| Britton [50] | USA | Not applicable | Skilled nurse residents | 463 | Retrospective cohort study | December 29, 2020—February 12, 2021 | The electronic medical record chart abstraction | BNT162b2 | SAR-CoV2 infection |
| Dagan [51] | Israel | Not stated | Healthcare workers | 4.7 million | Retrospective observational study | December 20, 2020—February 1, 2021 | Clallit Health Services (CHS) | BNT162b2 | SAR-CoV2 infection, symptomatic COVID-19 cases, severe COVID-19 cases, hospital admissions, deaths |
| Pritchard [39] | U.K. | Department of Health and Social Care, Welsh Government and Department of Health on behalf of the Northern Ireland Government and Scottish Government. | General population ≥16 years old | 383,812 | A large household survey with longitudinal follow-up | December 1, 2020—May 8, 2021 | The Office for National Statistics (ONS) COVID-19 Infection Survey | BNT162b2, ChAdOx1-S | SAR-CoV2 infection and infection severity |
| Domi [52] | USA | Not stated | Healthcare workers from CDC Tiberius system for Long Term Care facilities | 12,347 | Retrospective observational study | December 20, 2020—February 7, 2021 | The CMS National Health Safety Network (NHSN) Public File Data | BNT162b2 | SAR-CoV2 infection and mortality |

(*Continued*)

**Table 2.** (Continued)

| | Country | Funding source | Population | Sample size | Study design* | Study time frame | Database(s) | Type(s) of vaccine | Outcome |
|---|---|---|---|---|---|---|---|---|---|
| Jones [53] | U.K. | Wellcome Trust/Medical Research Council/NHS Blood and Transplant/ EPSRC | Healthcare workers | Approximately 9000 | Retrospective cohort study | January 18, 2021— January 31, 2021 | Hospital-laboratory interface software, Epic (Verona, WI) | BNT162b2 | SAR-CoV2 infection |
| Gras-Valenti [10] | Spain | Not stated | Healthcare workers in Alicante General Hospital | 268 | Test negative case control | January 25, 2021— February 7 2021 | Registro Nominal de Vacunas de la Generalitat Valenciana | BNT162b2 | SAR-CoV2 infection, symptomatic COVID-19 cases, |
| Jara [54] | Chile | The Agency Nacional de Investigacion & Millennium Science Initiative Program | Population ≥16 years old receiving at least 1 dose of CoronaVac | 10,187,720 | Prospective cohort study | February 2, 2021— May 1, 2021 | Database of Fondo Nacional de Salud (FONASA), the national public health insurance program. | CoronaVac | SAR-CoV2 infection, ICU admissions, deaths |
| Azamgarhi [55] | U.K. | Not stated | Healthcare workers in tertiary orthopaedic hospital in London | 1,409 | Retrospective cohort | January 15, 2021— March 26, 2021 | National Immunisation and Vaccination System (NIVS) | BNT162B2 | SAR-CoV2 infection |
| Knobel [56] | Spain | Not stated | Healthcare workers in Hospital del Mar in Barcelona, Spain | 2,462 | Screening method | December 1, 2021 – April 20, 2021 | Hospital del Mar administrative database | BNT162b2 | SAR-CoV2 infection |
| Harris [57] | England | Public Health England | General population from Household Transmission Evaluation Dataset (HOSTED) | 961 | Cohort study | January 4, 2021 – February 28, 2021 | Household Transmission Evaluation Dataset (HOSTED) and the National Immunization Management System (NIMS) | ChAdOx1 nCoV-19, BNT162b2 | SAR-CoV2 secondary infection |
| Zaqout [58] | Qatar | Qatar National Library | General population | 199,219 | Retrospective observational cohort | December 23, 2020— March 16, 2021 | The COVID-19 database at the Communicable Disease Center, Hamad Medical Corporation | BNT162b2 | SARS-CoV-2 infection |
| Mazagatos [59] | Spain | Not stated | Elderly aged 65 years and older | 338,145 | Cohort study | December 27, 2020–4 April 4,2021 | National Epidemiological Surveillance Network (RENAVE) and the National COVID-19 Vaccination Registry (REGVACU) | mRNA-1273 | SARS-CoV-2 infection |
| Abu-Raddad [29] | USA | Not stated | Population who received at least 1 dose of vaccine | 163,688 | Cohort study | March 8, 2021 -March 3, 2021 | The national federated Covid-19 databases | BNT162b2 | SARS-CoV-2 infection |

*(Continued)*

**Table 2.** (Continued)

| | Country | Funding source | Population | Sample size | Study design* | Study time frame | Database(s) | Type(s) of vaccine | Outcome |
|---|---|---|---|---|---|---|---|---|---|
| Flacco [31] | Italy | Not stated | General population aged 18 years old or older who were resident in the province of Pescara, Italy on 1 January 2021 | 245,226 | Retrospective cohort study | January 1, 2021— May 21, 2021 | Local Health Unit (LHU) of Pescara | BNT162b2, ChAdOx1 nCoV-19, mRNA-1273 | SARS-CoV-2 infection, hospitalisation, death |
| Kissling [32] | England, France, Ireland, the Netherlands, Portugal, Scotland, Spain and Sweden | European Union's Horizon | Population aged 65 years and older in primary care | 4,964 | Test-negative design | December 10, 2020— May 31, 2021 | I-MOVE-COVID-19 network | BNT162b2, ChAdOx1 nCoV-19 | SARS-CoV-2 infection |
| Thompson [60] | U.S.A. | National Center for Immunization and Respiratory Diseases and the Centers for Disease Control and Prevention | Healthcare workers | 3,975 | Prospective cohort study | December 14, 2020,- April 10, 2021 | Not applicable | BNT162b2, mRNA-1273 | SARS-CoV-2 infection |
| Kustin [30] | Israel | European Research Council (ERC) under the European Union's Horizon 2020 research and innovation programme, an Israeli Science Foundation and Milner and AppsFlyer foundations | Members of Clalit Health Services | 792 | Matched cohort study | January 23, 2021 to March 7, 2021 | CHS's data repositories | BNT162b2 | SARS-CoV-2 infection |
| Tang [61] | USA | American Lebanese Syrian Associated Charities (ALSAC) | Healthcare workers | 5,217 | Cohort study | December 17, 2020 -March 20, 2021 | St Jude Children's Research Hospital database | BNT162b2 | SARS-CoV-2 infection |
| Zacay [62] | Israel | Not stated | Member of The Meuhedet Health Maintenance Organization (MHMO) aged 16 years or older who had at least 2 PCR tests during November, at least 2 PCR tests during December, and at least 1 PCR test during January | 6,286 | Cohort study | January 1, 2021— February 11, 2021 | The Meuhedet Health Maintenance Organization (MHMO) | BNT162b2 | SARS-CoV-2 infection |

(*Continued*)

**Table 2.** (Continued)

| | Country | Funding source | Population | Sample size | Study design* | Study time frame | Database(s) | Type(s) of vaccine | Outcome |
|---|---|---|---|---|---|---|---|---|---|
| Jaiswal [28] | India | Not stated | Police personnel in Tamil Nadu | 117,524 | Real world data analysis | 13 Apr—14 May 2021 | Department of Police in Tamil Nadu database | ChAdOx | SARS-CoV-2 incidence of death |
| Garvey [63] | UK | Not stated | Healthcare workers at University Hospitals Birmingham (UHB) NHS Foundation Trust | 25,335 | Retrospective cohort | March 28, 2020 – March 21, 2021 | Occupational health database of all COVID-19 positive healthcare workers | BNT162b | SARS-CoV-2 infection |
| Walsh [64] | Ireland | The Clinical Governance Department at Beaumont Hospital | All permanently employed healthcare workers during the first 8 weeks of the staff vaccination programme in Ireland hospital | 4,458 | Cohort study | December 29, 2020 – February 22, 2021 | Hospital database | BNT162b2 | SARS-CoV-2 infection |
| Gupta [65] | US | US Department of Veterans Affairs | VA Boston Healthcare System (VABHS) clinical and nonclinical healthcare workers | 4,028 | Retrospective cohort | December 22, 2020 – February 1, 2021 | Not specified | mRNA-1273 | SARS-CoV-2 infection |
| Chodick [66] | Israel | Not stated | General population aged 16 and older who were vaccinated with at least one dose of the BNT162b2 vaccine during a mass immunization program from December 19, 2020—February 20, 2021 | 1,178,597 | Retrospective cohort study | December 19, 2020—March 3, 2021 | Maccabi Healthcare Services (MHS) database | BNT162b2 | SARS-CoV-2 infection, hospitalsation and mortality |

RT-PCR—reverse transcriptase polymerase chain reaction; MoH—Ministry of Health; ICU—intensive care unit; VE—vaccine effectiveness

*As reported in the study. For the purposes of standardisation in our analysis, we re-classified the following studies (in accordance with the WHO interim guidance for conducting vaccine effectiveness studies in LMICs): Tenforde et al—test negative case control design; Haas et al—retrospective cohort study; Sansone et al—retrospective cohort study; Keehner et al—retrospective cohort study; Thompson et al—retrospective cohort study; Benenson et al—screening study; Chodick et al—retrospective cohort study; Jameson et al—retrospective cohort study; Daniel et al—retrospective cohort study; Amit et al—retrospective cohort study; Dagan et al—prospective cohort study; Pritchard et al—prospective cohort study.

flowchart, although all studies were judged to be in a relevant population and setting. For the test-negative design case control studies, 2 studies were conducted in older adults [33, 41], whilst 2 studies were conducted in health workers ([9, 10]. However, 1 test-negative design case control study was in the general population [32], which may be subject to collider bias.

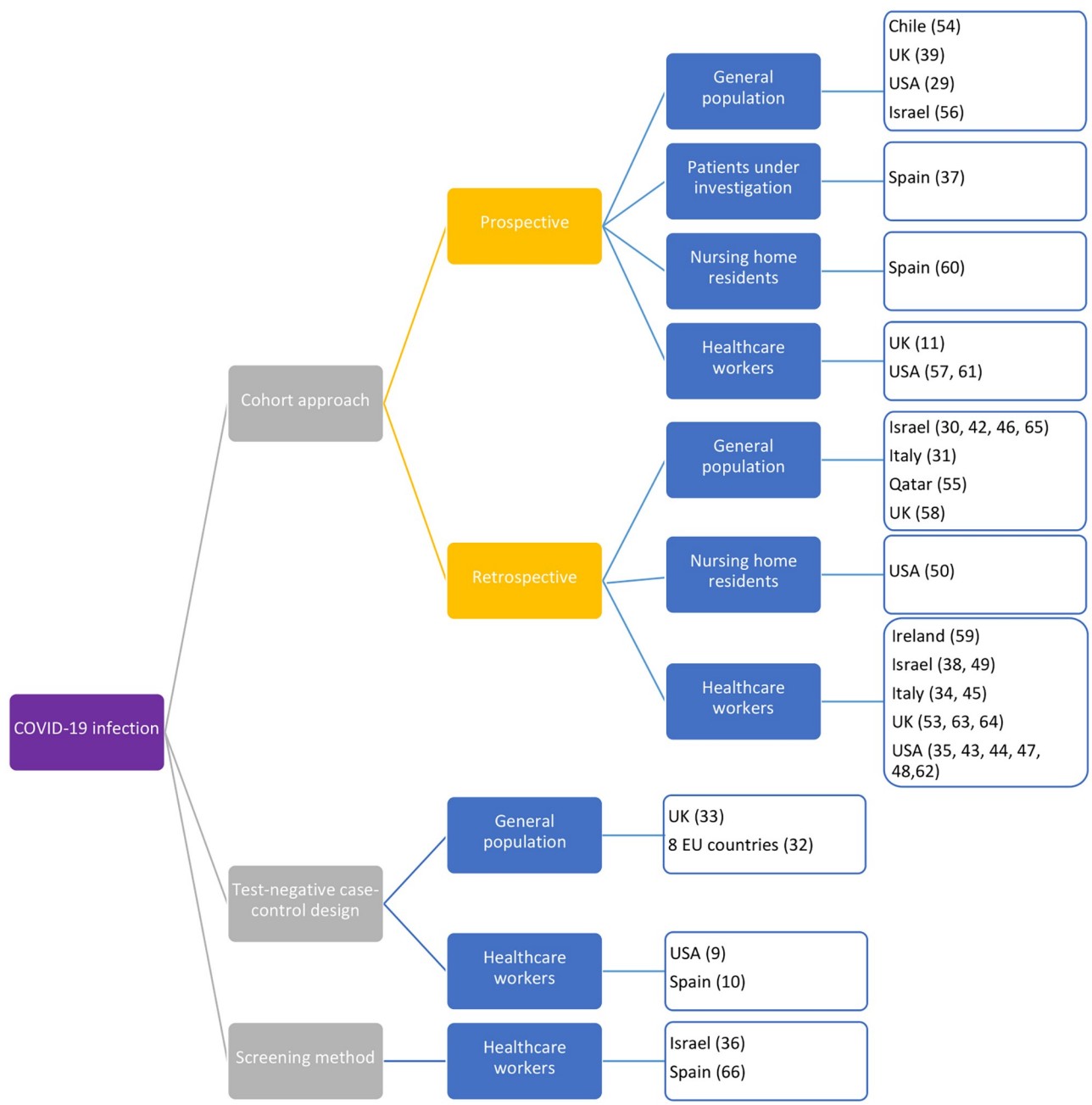

**Fig 2. Study design by outcome for COVID-19 vaccine effectiveness studies meeting inclusion criteria.**

Fourthly, due to the observational study design, selection bias and confounding effects were inevitable limitations. However, 22 studies did not report on assessment and mitigation of potential confounders (Fig 3). The most commonly reported confounders were age [9–11, 29–33, 37, 39–42, 44–46, 50, 51, 54–57, 59–62, 66], sex [9, 10, 29–33, 37, 39–42, 44–46, 50, 51, 54–57, 60–62, 66], socio-demographic factors (ethnicity/religion) [11, 33, 39, 41, 44, 50–52, 55, 60, 61, 66], geographical location [10, 11, 30, 33, 39, 41, 44, 51, 52, 54, 57, 62], chronic disease and/ or comorbidities [9, 11, 31, 32, 37, 39, 40, 50, 51, 54, 60, 66], time [10, 33, 36, 37, 40–42, 50, 52, 57], occupation [10, 11, 39, 44, 45, 55, 56, 60], and socio-economic status [33, 39, 40, 54, 57,

**Table 3. Methodology of included studies.**

| First author | Time of outcome assessment | Vaccination status assessment method | Method of handling outcome misclassification error | Were analyses restricted to symptomatic cases? | Types of biases and method of minimisation | Management of missing data | Management of potential confounder |
|---|---|---|---|---|---|---|---|
| Lopez-Bernal [33] | After 1st dose, 0–3, 4–6, 7–9, 10–13, 14–20, 21–27, 28–34, 35–41 and ≥ 42 days.; After 2nd dose, 0–3, 4–6, 7–13, and ≥ 14 days. For ChAdOx1-S the final interval was ≥35 days. | National Immunisation Management System | Not reported | No | Narrow follow-up windows (two periods each week up to 14 days and weekly thereafter) | Not reported | Possible confounders were included in the fully adjusted logistic regression model including age (in five year age groups, at 31 March 2021), sex, ethnicity, geography (NHS region), index of multiple deprivation, care home residence, and week of symptom onset. |
| Vasileiou [40] | 0–6, 7–13, 14–20, 21–27, 28–34, 35–41, and 42 or more days post-vaccination | Electronic health record data and national databases | Developed a national linked dataset and have created a platform that allowed rapid access to an analysis of data on vaccination status and medical condition status from routinely collected electronic health record data and national databases | No | Adjusted for time to adjust for any impact on the effect of these interventions and the course of the pandemic on estimates of vaccine effects | Separate group for individuals were created | Both the Cox models and Poisson regression used sampling weights to correct for the size of the registered general practice population being greater than the population in Scotland. Dementia was included as a functional variable to adjust for the residual confounding in which vaccines were not offered to or were declined by the most frail. Falsification of exposure sensitivity analysis assessed possible vaccine programme or residual confounding effects. |
| Tenforde [41] | Single-dose vaccinated less than 2 weeks before illness onset, defined as receipt of the first vaccine dose within 14 days before onset of COVID-like illness; 3) partially vaccinated, defined as receipt of 1 dose of a 2-dose vaccine series (Pfizer-BioNTech or Moderna) ≥14 days before illness onset or receipt of 2 doses, with the second dose received <14 days before illness onset; 4) fully vaccinated, defined as receipt of both doses of a 2-dose vaccine series, with the second dose received ≥14 days before illness onset | Self-report | Not reported | No | Self-reported data selection bias No minimization mentioned | Not reported | Not reported |

(*Continued*)

**Table 3.** (Continued)

| First author | Time of outcome assessment | Vaccination status assessment method | Method of handling outcome misclassification error | Were analyses restricted to symptomatic cases? | Types of biases and method of minimisation | Management of missing data | Management of potential confounder |
|---|---|---|---|---|---|---|---|
| Haas [42] | At least 7 days after second dose, ≥7 days after the second dose | National surveillance data | Exclude a small number of people who were initially reported to be asymptomatic but were later hospitalised for or died from COVID-19 | No | Israel's SARS-CoV-2 testing policy was different for unvaccinated and vaccinated individuals during the study period. At 7 days after the second dose, vaccinated individuals were exempt from the SARS-CoV-2 testing required of individuals who either had contact with a laboratory-confirmed case or returned from travel abroad. Some presymptomatic individuals who later developed symptoms without being hospitalised or dying might still have been included. No minimization mentioned. | Not reported | Multivariated and stratified analysis according to age groups |
| Sansone [34] | At least 7 days after 2nd dose | Hospital database | Not reported | Yes | Not reported | Not reported | Not reported |
| Keehner [43] | 1–7, 8–14 and 15 or longer | Electronic employee health record system | Not reported | No | Not reported | Not reported | Not reported |
| Thompson [44] | For unvaccinated person-days to partial immunization person-days: ≥14 days after first dose and before second dose. For full immunization person-days: ≥14 days after second dose | Self-report in electronic surveys, by telephone interviews, and through direct upload of vaccine card images at all sites; electronic medical records | Not reported | No | Not reported | Not reported | Not reported |
| Fabiani [45] | Between 14–21 days after the administration of the first dose; between at least 7 days after the administration of the second dose | Local COVID19 surveillance database | Not reported | No | Not reported | Not reported | Adjusted for potential confounders based on available data |
| Cavanaugh [35] | Within 7 days | Immunization registry review and facility interviews | Not reported | Yes | Not reported | Not reported | Not reported |

*(Continued)*

**Table 3.** (*Continued*)

| First author | Time of outcome assessment | Vaccination status assessment method | Method of handling outcome misclassification error | Were analyses restricted to symptomatic cases? | Types of biases and method of minimisation | Management of missing data | Management of potential confounder |
|---|---|---|---|---|---|---|---|
| Hall [11] | symptomatic testing was done at any time during the presentation of symptoms | Registry of COVID-19 vaccination in England | Not reported | No | Defined the end of follow-up in none-positive cases as the date of a negative test, if the test was after this date, to avoid immortal time bias. Vaccinated population had slightly higher testing frequency than the unvaccinated population and therefore it was more likely to pick up infections among the vaccinated, resulting in biasing vaccine effectiveness results towards the null hypothesis. Possibility of recall bias due to Self-completed questionnaires but should not have affected symptom reports by vaccination status. Excluded differential symptom reporting. Healthy worker effect bias might underestimate the disease impact compared with the general population | The follow-up time was censored at the date of the suspect second dose if a participant had an unreliable date of a second dose (eg, a second dose administered before a first dose or administered less than 19 days after the first dose) | Full model was adjusted for site as a random effect, period, and eight fixed effects: age, gender, ethnicity, comorbidities, job role, frequency of contact with COVID-19 patients, employed in a patient facing role, and occupational exposure |
| Benenson [36] | All time points and compare incidence rate between weeks after vaccination | Human Resources Department | Not reported | No | No mandated PCR screenings after second dose of vaccination, leading to underdiagnosed COVID-19 but HCWs were tested following every mild symptom and following exposure to previously unknown patients or colleagues | Not reported | Not reported |
| Martínez-Baz [37] | >14 days after first dose | Navarre Health Service | 2 days before the onset of symptoms in the case to 10 days after the onset of symptoms, or in the 2 days before the sample; 10 days after the sample was taken for asymptomatic cases | No | As close contacts of COVID-19 cases have had a known risk exposure, comparison between vaccinated and unvaccinated close contacts is an ideal design | Not reported | Not reported |

(*Continued*)

**Table 3.** (Continued)

| First author | Time of outcome assessment | Vaccination status assessment method | Method of handling outcome misclassification error | Were analyses restricted to symptomatic cases? | Types of biases and method of minimisation | Management of missing data | Management of potential confounder |
|---|---|---|---|---|---|---|---|
| Chodick [46] | Daily and cumulative infection rates in days 13 to 24 were compared with days 1 to 12 after the first dose | Central databases of Maccabi Healthcare Services (MHS), Health Maintenance Organization (HMO) in Israel | Limiting the analysis to infections with documented COVID-19 symptoms; calculated cumulative incidence of infection during a 12-day period (days 13–24 after first dose) compared with days 1 to 12 after vaccination with the first dose; excluded positive PCR prior to the index date and those who joined MHS after February 2020 (incomplete medical history) | Yes | Minimal information bias due to automated data collection of vaccination status and laboratory results that are offered to all citizens free of charge. Minimal selection and indication bias from comparing vaccinated versus unvaccinated or test-negative studies and comparing vaccinated individuals in different time intervals after immunization. More asymptomatic infections may go undocumented because change in health seeking behavior and decreased test rate 2 weeks after first dose. | Did not censor the follow-up period at date of second dose to avoid a potential selection-bias because individuals with a positive SARS-CoV-2 test result after the first dose are recommended to postpone their second dose. | Not reported |
| Jameson [47] | For full immunization: 7 days after second dose | Not reported | Not reported | Yes | Voluntary nature of the vaccine program is to select individuals at decreased risk of COVID-19 acquisition regardless of vaccination and possibility of detecting ongoing shedding from a remote infection, might only test symptomatic. | Not reported | Not reported |
| Pilishvili [9] | Effectiveness of a single dose was measured during the interval from 14 days after the first dose through 6 days after the second dose; exclude participants tested within 0–2 days of receiving the second dose; effectiveness of 2 doses was measured ≥7 days after the receipt of the second dose | Occupational health or other verified sources (e.g., vaccine card, state registry, or medical record). | Daily screening for symptoms of COVID-19: referred to complete nasopharyngeal swab testing for COVID-19 before returning to work | Yes | Testing was based on occupational health practices at each facility, and no changes in routine testing practices were reported after vaccine introduction. | Not reported | Not reported |
| Daniel [48] | Partially vaccinated: one dose or ≤ 7 days post-second dose BNT162b2 vaccination or ≤ 14 days post-second dose mRNA-1273 vaccination; fully vaccinated: ≥ 7 days post-second dose BNT162b2 vaccination or ≥ 14 days post-second dose mRNA-1273 vaccination | Vaccination record from University of Texas Southwestern Medical Center (UTSW) | Not reported | No | Not reported | N/A | N/A |

(*Continued*)

**Table 3.** (Continued)

| First author | Time of outcome assessment | Vaccination status assessment method | Method of handling outcome misclassification error | Were analyses restricted to symptomatic cases? | Types of biases and method of minimisation | Management of missing data | Management of potential confounder |
|---|---|---|---|---|---|---|---|
| Angel [38] | Days 7–28 after first dose (partially vaccinated); and >21 days after second dose (fully vaccinaetd); Median follow-up time = 63 days (Dec 20, 2020, to Feb 25, 2021) | Employee health database | Not reported | No | Two groups may not be comparable, which is minimized by using used propensity score matching Vaccinated group had fewer tests. | Not reported | Regression models were used to adjust confounders. Other confounders may be present that were unaccounted for in the regression analyses and in the adjustments for propensity score |
| Amit [49] | Days 1–14 and 15–28 after the first dose of the vaccine | Medical center's database | Not reported | No | Lack of active laboratory surveillance in the cohort might have resulted in an underestimation of asymptomatic cases. | Not reported | Rate ratio of new cases in vaccinated compared with unvaccinated HCWs each day were adjusted for community exposure rates using Poisson regression. |
| Britton [50] | Partially vaccinated (>day 14 after first dose through day 7 after second dose); fully vaccinated (>7 days after second dose); Started on the date of first vaccination clinic (December 29, 2020 for facility A and December 21, 2020 for facility B) and ended on February 9, 2021 and February 12, 2021, respectively | Electronic chart review | Not reported | Yes | Not reported | The ethnicity could not be reported because ethnicity data were missing for 30% of residents | Not reported |
| Dagan [51] | Days 14 through 20 after the first dose of vaccine; days 21 through 27 after the first dose (administration of the second dose was scheduled to occur on day 21 after the first dose); day 7 after the second dose until the end of the follow-up | Clalit Health Services (CHS) database, the largest of four integrated health care organizations in Israel, | Not reported | No | To assess a possible selection bias that could stem from informative censoring, whereby controls who are vaccinated feel well around the time of vaccination and sensitivity analysis was performed in which they were kept in the unvaccinated group for a period of time that was set differently for each outcome. | The date of onset of symptoms was not available for the analysis and the date was set to the date of swab collection for the first positive PCR test. | Performed rigorous matching on a wide range of factors that may be expected to confound the causal effect of the vaccine on the various outcomes. Population groups with high internal variability in the probability of vaccination or outcome were excluded, such as health care workers, persons confined to the home for medical reasons, and nursing home residents, to avoid residual confounding; |
| Pritchard [39] | ≥21 days after the first dose and post-second dose | Self-reported | Not reported | Yes | This study was designed as a large-scale community survey recruiting from randomly selected private residential households, providing a representative sample of the UK general population; | Not reported | Unbiased sampling frame, which exploited for our logistic regression rather than having to censor individuals. |

*(Continued)*

**Table 3.** (Continued)

| First author | Time of outcome assessment | Vaccination status assessment method | Method of handling outcome misclassification error | Were analyses restricted to symptomatic cases? | Types of biases and method of minimisation | Management of missing data | Management of potential confounder |
|---|---|---|---|---|---|---|---|
| Domi [52] | Four time-dependent, delayed vaccination effects at3, 4, 5, and 6 weeks respectively after the first vaccination. | Data registry: National Health Safety Network (NHSN) Public File data | Not reported | No | Not reported | Not reported | To address the highly skewed, longitudinal countmeasurements with a large proportion of zeros. The negative binomial model addresses the issue of overdispersion by including a dispersion parameter that relaxes the assumption of equal mean and variance of the Poisson model. |
| Jones [53] | ≥12 days post-vaccination | Hospital data registry: Cambridge University Hospitals NHS Foundation Trust (CUHNFT) | This study was used real-time RT-PCR, with all sample processing and analysis undertaken at the Cambridge COVID-19 Testing Centre (Lighthouse Laboratory). | Yes | The date of infection could have been earlier than the test date, may lead to an underestimate of the vaccine's effect (bias towards the null). | Not reported | Not reported |
| Gras-Valenti [10] | After 12 days after the first dose | Hospital data registry | The determinationtion of SARS-CoV-2 in an aspiration sample nasopharyngeal tract during the first 24 hours after patient's consultation. If negative, they were follow-up and another PCR was repeated at tendays of the last contact with the case. | Yes | Not reported | Not reported | Variables that showed statistically significant differences between vaccinated and non-vaccinated HW were included in the regression model. |
| Jara [54] | Partial immunization (≥14 days after receipt of the first dose and before receipt of the second dose) and full immunization (≥14 days after receipt of the second dose) | National data registry | Those periods in this study were excluded from the at-risk person-time in our analyses. | Yes | Sub-group analysis to investigate healthcare access between RT-PCR and antigen testing, and between 16–59 years and adults over 60 years | Not reported | This study was evaluated the robustness of the model assumptions by fitting a stratified version of the extended Cox proportional-hazards model. |
| Azamgarhi [55] | > 10 days after vaccination | Registry | Not reported | No | Missing data about vaccine information, inclusion of potentially less susceptible individuals in the unvaccinated arm would be to make the vaccine appear more effective. | Significant efforts were made to obtain data on HCWs that received the vaccine elsewehere. | Groups were compared adjusting for demographic details found to vary significantly between groups. Hazard ratios were also adjusted for underlying COVID-19 infection rates in the London area. |
| Knobel [56] | 2 weeks after the first dose and 1 week after the second dose | Hospital database | Not reported | No | Prone to random error due to small number of outcomes. | Not reported | Not reported |

(*Continued*)

**Table 3.** (Continued)

| First author | Time of outcome assessment | Vaccination status assessment method | Method of handling outcome misclassification error | Were analyses restricted to symptomatic cases? | Types of biases and method of minimisation | Management of missing data | Management of potential confounder |
|---|---|---|---|---|---|---|---|
| Harris [57] | Vaccinated 21 days or more prior to testing positive for COVID-19 | National Immunisation Management System | Not reported | No | Bias could occur if case ascertainment differed between household contacts of vaccinated persons and those of unvaccinated persons; no method of minimisation | Not reported | Logistic-regression models were used to adjust for the age and sex of the person with the index case of Covid-19 (index patient) and the household contact, geographic region, calendar week of the index case, deprivation (a composite score of socioeconomic and other factors), and household type and size. Timing of effects among index patients who had been vaccinated at any time up to the date of the positive test was also considered. |
| Zaqout [58] | During days 1–7, 8–14,15–21, 22–28, and >28 days post-vaccination | Clinical data registry | Not reported | Yes | Not reported. | Not reported | Not reported |
| Mazagatos [59] | Partially vaccinated—dose 1: Vaccinated with the first dose of Comirnaty or Moderna COVID-19 vaccine, and more than 14 days since vaccination. Partially vaccinated—dose 2: Vaccinated with two doses of Comirnaty or Moderna COVID-19 vaccine, and less than 7 days since the second dose for Comirnaty or less than 14 days for Moderna COVID-19 vaccine. Full immunity not reached. Fully vaccinated: Vaccinated with two doses, and 7 days or more after the second dose for Comirnaty and 14 days or more for Moderna COVID-19 vaccine. Full immunity reached. | Vaccination status were retrived from the National COVID-19 Vaccination Registry (REGVACU). | Not reported | No | Not reported | Not reported | Not reported |
| Abu-Raddad [29] | N/A (any) | Standardized national SARS-CoV-2 database | Not reported | No | Test negative case control design to control for bias that may result from differences in health care–seeking behavior between vaccinated and unvaccinated persons | Not reported | Two sensitivity analyses were conducted by first matching by the exact testing date and second by a logistic regression to adjust for calendar week |
| Flacco [31] | 14 days after the second dose for all vaccines | Registry | Not reported | No | Recall or misclassification bias of vaccination status | Not reported | Not reported |

(*Continued*)

**Table 3.** (Continued)

| First author | Time of outcome assessment | Vaccination status assessment method | Method of handling outcome misclassification error | Were analyses restricted to symptomatic cases? | Types of biases and method of minimisation | Management of missing data | Management of potential confounder |
|---|---|---|---|---|---|---|---|
| Kissling [32] | > = 14 days post vaccination | Not mentioned | Not mentioned | Yes | Tested sampling bias with phylogenetic tree | Imputed study sites where date of symptom onset was not available (one site) or had more than 25% of missing information (two sites) as 3 days before the swab date (3 days was the median delay between onset and swab in the pooled data). | Not reported |
| Thompson [60] | Fully vaccinated (≥14 days after dose 2), partially vaccinated (≥14 days after dose 1 and <14 days after dose 2), or unvaccinated or to have indeterminate vaccination status (<14 days after dose 1) | Self-assessed electronic and telephone surveys, direct upload of images of vaccination cards and electronic medical records, occupational health records, or state immunization registries were reviewed at the sites in Minnesota, Oregon, Texas, and Utah | A sensitivity analysis removed person-days when participants had possible misclassification of vaccination status | No | Selection biases was minimised by stratifying recruitment of participants according to site, sex, age group, and occupation; Recall and confirmation biases due to that results for febrile symptoms and duration of illness were based on participant-reported data, minimized by comparing these findings with the virologic findings of a reduced viral RNA load and duration of viral RNA detection among vaccinated participants. | Not reported | Use of an inverse probability of treatment weighting approach. Generalized boosted regression trees were used to estimate individual propensities to be at least partially vaccinated during each study week, on the basis of baseline sociodemographic and health characteristics and the most recent reports of potential virus exposure and PPE use. |
| Kustin [30] | Controls who were not vaccinated before the positive PCR result. The dose1 group: individuals who had a positive PCR test that was performed between 14 days after the first dose and 6 days after the second dose. The dose2 group: and individuals who had a positive PCR test that was performed at least 7 days after the second vaccine dose. | Clalit Health Services databae. | Following classification by Pangolin, the authors noted that one dose1 control sequence, originally classified as WT (B.1.235), was located within the B.1.351 clade on the phylogenetic tree. Its pair was classified as B.1.1.7, and they included this pair in an extreme scenarios analysis. This is in line with recent concerns regarding misclassifications of Pangolin, and led to manually verify the phylogenetic location of all sequences in this study. | No | A phylogenetic tree of all the sequenced samples together with additional available sequences from Israel was reconstructed to test bias in sampling scheme and observed that vaccinated and unvaccinated samples were highly interspersed along the tree, ruling out strong biases in sampling. | Not reported | A conditional logistic regression was used as a sensitivity analysis to include age as a possible confounder in case that matching was not sufficient. |
| Tang [61] | Not mentioned | Not mentioned | Not reported | No | Not reported | Not reported | Not reported |
| Zacay [62] | 1. ≥14 day after the 1st dose 2. 1–6 days after the 2nd dose 3. ≥7 day after the 2nd dose | Health maintenance organiation (HMO) database | Not reported | No | Number of PCR tests varied across sub-groups. No method of minimization. | Not reported | Different rates of infection across sectors and calculated infection rates separately for each sector |

(*Continued*)

**Table 3.** (Continued)

| First author | Time of outcome assessment | Vaccination status assessment method | Method of handling outcome misclassification error | Were analyses restricted to symptomatic cases? | Types of biases and method of minimisation | Management of missing data | Management of potential confounder |
|---|---|---|---|---|---|---|---|
| Jaiswal [28] | Not mentioned | The Tamil Nadu Police department has been documenting vaccination of its workforce. | Not reported | Yes | Not reported | Not reported | No adjustment for potential confounders including age, comorbidities and previous exposure to COVID-19 infection could, as the vaccination details were collected as aggregated numbers. |
| Garvey [63] | > 10 days after vaccination | Registry | Not reported | No | Not reported | Not reported | Not reported |
| Walsh [64] | 0–7 days, 8–14 days, 15–21 days, 22–30 days, 39 days | Not reported | Not reported | No | Not reported | Not reported | Not reported |
| Gupta [65] | VE were measured before 8 and 15 days following the first dose of vaccination | Not reported | Not reported | No | Not reported | Not reported | Not reported |
| Chodick [66] | days 7–27 after the second dose | Registry | Not reported | No | More asymptomatic infections undocumented but this potential information bias is likely insignificant, as VE calculated for all infections was similar or lower to the one calculated for symptomatic cases | Not reported | Not reported |

**Fig 3. Quality assessment of included studies using the Real Life Evidence AssessmeNt Tool (RELEVANT).**

66]. Methods reported to manage confounders include adjusted logistic regression model [10, 11, 29, 30, 38, 45, 57, 60], stratified analysis [42, 54], matching cases and controls [51], and excluding population groups with high variability in the probability of vaccination or outcome [51]. 4 studies reported adjusting for or conducting sensitivity analysis by different exposure or infection rates [40, 49, 55, 62]. No study in our review measured adherence to NPIs and none of the test-negative design studies measured respiratory viral infection, which could bias likelihood of individuals seeking COVID-19 tests. Previous SARS-CoV-2 infection was not measured (or not reported) in the majority of studies, participants with prior infection were excluded in 16 studies, and 2 studies included prior infection in sensitivity analysis [10, 33]. Finally, only 14 of 26 studies reported on the extent of missing data (Fig 3). Studies reported dealing with missing data by creating a separate group for individuals with missing data [40], not including missing variables in the analysis [50, 60], or by mean imputation [32].

## Discussion

To our knowledge, this is the first systematic review of methodologies for COVID-19 vaccine effectiveness studies. Given the scale of COVID-19 vaccine roll-out thus far, our review identified relatively few studies assessing real-world vaccine effectiveness. All studies identified are from HICs, often utilising national databases (which may not exist or may be of poorer quality in LMICs), and the great majority assessed mRNA vaccines, which are more prevalent in HICs but only represent a third of the vaccines with WHO Emergency Use Listing (EUL) [67] and one-fifth of COVAX secured supply from legally binding agreements [68]. Whilst the WHO landscape of observational studies has identified pre-prints and registered studies being conducted in six middle income countries (Argentina, Brazil, India, Indonesia, Tunisia, Turkey) [18], between our review and the WHO landscape document there are few real-world effectiveness studies for vaccines that have received WHO EUL and no study in low-income countries. These findings underscore the importance of advocating for real-world effectiveness studies on all approved COVID-19 vaccines and across diverse LMIC settings.

Our review has highlighted several important components to consider at the outset of designing a real-world effectiveness study of COVID-19 vaccines, including the appropriate study design, study population, outcome, and time for follow-up. The most common study design identified in our review was a cohort approach, which may have been facilitated by the presence of large, reliable, and inter-linked databases in study countries. Test negative design case control studies were the second most common study design, but we did not identify any case-control studies in this review. We hypothesise that this finding may be because of the challenges in enrolling an unbiased comparison group: the low number of case-control registered studies and pre-prints suggests that we did not select against case-control studies by restricting our search to peer-reviewed articles [18].

In studies assessing symptomatic or asymptomatic infection as an outcome, healthcare workers were the most common study population. In many studies, healthcare workers were an opportune population due to routine symptomatic or RT-PCR screening activities undertaken within the health system. Conversely, we identified no studies using healthcare workers as the study population for the outcomes hospitalisation and death, which we hypothesise as being due to the low number of severe outcomes in this group [69]. Instead, studies either selected populations at high risk of disease (such as the elderly) or utilised large national databases to assess outcomes in the general population. If large-scale studies are not feasible, or rely on poor-quality databases, LMICs may find that test-negative designs are most feasible to implement, as recommended by the WHO interim guidance [13]. Regarding study population and outcome, we suggest that health workers may be the most appropriate population for

studies measuring effectiveness against infection, whereas studies on hospitalisation/death may best focus on elderly populations or other high risk groups.

Given the short timeline since COVID-19 vaccine introduction, the duration of all studies was less than five months. As would be expected, studies looking at hospitalisation and death tended to have longer duration than those assessing infection. However, the short follow-up time may have underestimated vaccine effectiveness against severe outcomes, and means that studies were not able to consider duration of protection, which will be important in informing strategies for delivering booster doses among different populations. Studies of longer duration may also allow assessment of changing vaccine effectiveness with the emergence of new VOCs. Despite widespread concern on protection of COVID-19 vaccines against VOCs, many studies did not assess prevalence of variants and none reported on the delta strain. The WHO landscape of observational studies for vaccine effectiveness suggests that this is likely to remain a significant gap in the literature for future research to consider [18].

Our review highlights several gaps that merit further study, alongside opportunities to strengthen the quality of real-world vaccine effectiveness studies. Firstly, we identified a need for studies in LMICs, especially in Africa and Asia, as well as effectiveness studies with a longer duration and covering all vaccines with WHO EUL. Without information on vaccine effectiveness for all licensed products, governments may face diminishing public confidence towards the vaccines in use in their country. Second, most studies did not calculate (or report) the sample size a priori. Whilst this may be less relevant for retrospective cohort studies based on national databases, which often utilise thousands or millions of records, it is an important consideration for prospective study designs or smaller scale retrospective cohort studies. Since many LMICs are unlikely to be able to replicate the large-scale studies from HICs, calculating minimum sample size will be very important, and should account for differences in access to healthcare services and health seeking behaviour in LMICs, as compared with HICs. Third, we identified weaknesses across studies in identifying and mitigating against potential confounders, and in reporting on missing data. Missing data are likely to be a greater issue in LMICs and differences in healthcare utilisation are likely to be more pronounced than in many HICs, requiring a well-considered plan for identifying and dealing with confounders and missing data. In particular, we note that many studies either did not measure for previous SARS-CoV-2 infection or used this as an exclusion criterion. If the infrastructure exists, we recommend testing for previous infection and conducting sensitivity analysis including this group, to avoid selecting the sample based on exposure risk. Finally, most studies failed to report on the presence of VOCs or on conflict of interest, including funding source. The former is important to respond to changes in vaccine effectiveness with new variants, and the latter is important for credibility of studies for policymaking. Accordingly, we recommend a number of additions to the WHO interim guidance on evaluation of COVID-19 vaccine effectiveness. The document would benefit from further guidance on setting an appropriate time horizon for studies, alongside guidance on designing studies that can be conducted with limited resources. We also propose the inclusion of practical guidance on identifying important confounders for a given setting and management of missing data. Finally, we suggest the inclusion of managing and reporting conflict of interest, as a fundamental part of study design.

There are several limitations to our review. We conducted the review only seven months after the first COVID-19 vaccines were licensed, limiting the number of studies and timeframe, as well as skewing our search results towards HICs, which were the first to introduce COVID-19 vaccination. Restricting our search to peer-reviewed articles further limited the number of results and favoured earlier studies in HICs with limited outcomes based on available data. Because of these limitations, our review was unable to objectively compare approaches that may be more appropriate to LMIC settings. Furthermore, because of an

urgent request from the Thai government, we employed rapid review methodology. Consultation with experts identified six additional papers that were not captured by our search terms, and there may be other studies which we missed. However, because the focus of our review is methodology of studies and not an estimate of vaccine effectiveness, we believe that this is acceptable. Particularly for the quality assessment of studies, we had to make assumptions based on reporting in the article, whereas contacting study authors for clarifications may have yielded further information to enhance our analysis.

Despite the importance of real-world effectiveness studies for informing national COVID-19 prevention and control policies in LMICs, existing studies tend to focus on settings, available vaccines, and VOCs specific to a handful of HICs. Although WHO recommends against conducting effectiveness studies in each country [13], in light of the heterogeneity between studies, we argue that there is benefit to each country designing and conducting effectiveness studies, subject to available resources. Considerable funding has been made available from the public sector for COVID-19 vaccine development and deployment. We therefore argue that it is imperative for the public sector to continue funding to the end of the product development continuum and finance studies on effectiveness and impact, not just domestically but across countries, given the global nature of the COVID-19 pandemic.

In summary, our review highlights the importance of local vaccine effectiveness data, and in providing further guidance on important confounders and methods for managing missing data. Most vaccine effectiveness studies to date have been conducted in HICs with access to reliable and interlinked databases for COVID-19 vaccination, diagnosis and treatment. Such databases often do not exist in LMICs, meaning that countries will be employing prospective study designs, requiring a priori calculation of sample size and a clear plan to manage and report on confounders and missing data. We highlight the limited experience conducting vaccine effectiveness in LMICs, but emphasise the importance of such studies for policymakers in LMICs to develop and monitor vaccination policies, as well as to enhance public confidence in vaccination. We call on the global community to support LMICs to lead and implement COVID-19 vaccine effectiveness studies in their settings, as a priority research area moving forward.

## Supporting information

**S1 Appendix. Search strategy and list of articles excluded at full text screening.**
(DOCX)

**S2 Appendix. Data extraction form.**
(XLSX)

**S1 Checklist.**
(DOCX)

## Acknowledgments

The authors would like to acknowledge Vice Public Health Minister Sopon Mekthon who commissioned this study. The Health Intervention and Technology Assessment Program (HITAP) is supported by the International Decision Support Initiative (iDSI) to provide technical assistance on health intervention and technology assessment to governments in low- and middle-income countries. iDSI is funded by the Bill & Melinda Gates Foundation, the UK's Department for International Development, and the Rockefeller Foundation. HITAP is also supported by the Access and Delivery Partnership, which is hosted by the United Nations Development Programme and funded by the Government of Japan.

## Author Contributions

**Conceptualization:** Yot Teerawattananon.

**Data curation:** Thunyarat Anothaisintawee, Nuttakarn Budtarad.

**Formal analysis:** Yot Teerawattananon, Thunyarat Anothaisintawee, Chatkamol Pheerapanyawaranun, Siobhan Botwright, Katika Akksilp, Natchalaikorn Sirichumroonwit, Nuttakarn Budtarad, Wanrudee Isaranuwatchai.

**Funding acquisition:** Yot Teerawattananon, Wanrudee Isaranuwatchai.

**Investigation:** Nuttakarn Budtarad.

**Methodology:** Yot Teerawattananon, Thunyarat Anothaisintawee, Chatkamol Pheerapanyawaranun, Katika Akksilp, Wanrudee Isaranuwatchai.

**Project administration:** Yot Teerawattananon, Siobhan Botwright.

**Supervision:** Yot Teerawattananon, Thunyarat Anothaisintawee, Wanrudee Isaranuwatchai.

**Validation:** Yot Teerawattananon, Thunyarat Anothaisintawee, Chatkamol Pheerapanyawaranun, Siobhan Botwright, Katika Akksilp, Natchalaikorn Sirichumroonwit, Wanrudee Isaranuwatchai.

**Visualization:** Chatkamol Pheerapanyawaranun, Natchalaikorn Sirichumroonwit.

**Writing – original draft:** Yot Teerawattananon, Thunyarat Anothaisintawee, Siobhan Botwright.

**Writing – review & editing:** Yot Teerawattananon, Wanrudee Isaranuwatchai.

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
