## [Decision Letter · Decision Letter 0]

1 Oct 2021

PONE-D-21-26919A systematic review of methodological approaches for evaluating real-world effectiveness of COVID-19 vaccines: advising resource-constrained settingsPLOS ONE

Dear Dr. Botwright,

Thank you for submitting your manuscript to PLOS ONE. After careful consideration, we feel that it has merit but does not fully meet PLOS ONE’s publication criteria as it currently stands. Therefore, we invite you to submit a revised version of the manuscript that addresses the points raised during the review process.

We look forward to receiving your revised manuscript.

Kind regards,

Chaisiri Angkurawaranon

Academic Editor

PLOS ONE

Journal Requirements:

“This study was funded by the Health Systems Research Institute, grant number 64134002RM011L0. The funder of the study had no role in study design, data collection, data analysis, data interpretation, or writing of the report. The authors would like to acknowledge Vice Public Health Minister Sopon Mekthon who commissioned this study and Ms.Nuttakarn Budtarad who help on protocol development and abstract screening. The Health Intervention and Technology Assessment Program (HITAP) is supported by the International Decision Support Initiative (iDSI) to provide technical assistance on health intervention and technology assessment to governments in low- and middle-income countries. iDSI is funded by the Bill & Melinda Gates Foundation (OPP1202541), the UK’s Department for International Development, and the Rockefeller Foundation. HITAP is also supported by the Access and Delivery Partnership”

We note that you have provided funding information within the Acknowledgements Section. Please note that funding information should not appear in the Acknowledgments section or other areas of your manuscript. We will only publish funding information present in the Funding Statement section of the online submission form.

“This study was funded by the Health Systems Research Institute (https://hsri.or.th/researcher), grant number 64134002RM011L0.

The funder of the study had no role in study design, data collection, data analysis, data interpretation, or writing of the report.”

Reviewers' comments:

Reviewer's Responses to Questions

**Comments to the Author**

1. Is the manuscript technically sound, and do the data support the conclusions?

Reviewer #1: Partly

Reviewer #2: No

2. Has the statistical analysis been performed appropriately and rigorously? 

Reviewer #1: N/A

Reviewer #2: N/A

3. Have the authors made all data underlying the findings in their manuscript fully available?

Reviewer #1: Yes

Reviewer #2: Yes

4. Is the manuscript presented in an intelligible fashion and written in standard English?

Reviewer #1: Yes

Reviewer #2: Yes

5. Review Comments to the Author

Reviewer #1: In this rapid review, Teerawattananon and colleagues summarize methodological approaches for evaluating real-world effectiveness of COVID-19 vaccines, with a particular focus on the need for these types of studies in low- and middle-income countries (LMICs).

As a general comment, the data from this review are under-analyzed. Most presented results are unrelated to stated research objective (to summarise methodological approaches being used to study real-world COVID-19 vaccine effectiveness). As the focus of this review is on methodological approaches, further details on study methodology should be summarized in the tables and text.

Figure 3, in particular, is misleading and not a stated objective of this rapid review. VE estimates will vary depending on study design, population, time period, etc. A critical assessment of the methodological biases present in these studies has not been provided on a study-by-study basis, which would allow for appropriate interpretation of these VE estimates. The focus of Results and Discussion should instead be on Figure 4. Additional methodological details should be added to Tables 1 and 2. The Introduction should focus more on methodological issues in conducting real-world VE studies, rather than a general discussion of COVID-19 vaccination strategies.

Finally, it is unclear what this systematic review adds to the body of literature. As mentioned in this review, the WHO recently published interim guidance on conducting VE studies in LMIC, which recommends the test-negative design as the most efficient and feasible method for LMICs. It also already summarizes key features of COVID-19 VE studies in its ‘Landscape of observational study designs on the effectiveness of COVID-19 vaccination’ evergreen document.

Specific comments:

1. Please update search strategy. Several recently published COVID-19 VE studies are missing from your review. The WHO landscape of observational studies document includes 90+ published studies (140+ when preprints included) where the main study outcome is ‘effectiveness’ but your review only includes 26 studies. Examples of missing citations (not a complete list):

a. Andrejko et al. Prevention of COVID-19 by mRNA-based vaccines within the general population of California. Clin Infect Dis. 2021 Jul 20. https://doi.org/10.1093/cid/ciab640

b. Chung et al. Effectiveness of BNT162b2 and mRNA-1273 covid-19 vaccines against symptomatic SARS-CoV-2 infection and severe covid-19 outcomes in Ontario, Canada: test negative design study. BMJ. 2021 Aug 10. https://doi.org/10.1136/bmj.n1943.

c. Pawlowski et al. FDA-authorized COVID-19 vaccines are effective per real-world evidence synthesized across a multi-state health system. Med (N Y). 2021 Jun 29. https://doi.org/10.1016/j.medj.2021.06.007.

d. Monge et al. Direct and Indirect Effectiveness of mRNA Vaccination against Severe Acute Respiratory Syndrome Coronavirus 2 in Long-Term Care Facilities, Spain. Emerg Infect Dis. 2021 Jul 27. https://doi.org/10.3201/eid2710.211184.

e. Shrotri et al. Vaccine effectiveness of the first dose of ChAdOx1 nCoV-19 and BNT162b2 against SARS-CoV-2 infection in residents of long-term care facilities in England (VIVALDI): a prospective cohort study. Lancet Infect Dis. 2021 Jun 23. https://doi.org/10.1016/S1473-3099(21)00289-9.

2. Methods. Were preprints included? Given the rapidly changing COVID-19 vaccine landscape, inclusion of only published, peer-reviewed articles in your rapid review may introduce a publication bias. This may explain in part the study characteristics presented in Table 2, with more VE studies published in certain HICs (e.g. Israel, UK, US), population groups that were prioritized for early vaccination (e.g. nursing home residents, healthcare workers) or earlier variants (e.g. Alpha).

3. Methods, page 4 “If effectiveness data were unclear, the study was not included in the comparison of effectiveness but was kept for the qualitative analysis of study design and methods.” Please clarify in what way the effectiveness data were unclear. Would these studies not be excluded based on your requirement for “peer-reviewed literature, as quality assurance for study design and reporting”?

4. Methods, page 4, “For studies reporting incidence rate ratio (IRR), the formula (1-IRR)*100 was used to calculate vaccine effectiveness.” What about studies reporting other effect estimates (e.g. odds ratio)?

5. Methods, page 4. The authors used the RELEVANT tool to asses study quality. While this may be a useful tool for reporting of observational studies, other tools (e.g. ROBINS-I: https://methods.cochrane.org/bias/risk-bias-non-randomized-studies-interventions) may be better suited to assess the risk of bias.

6. Methods, page 4, “Qualitative analysis identified areas of limited evidence and highlighted opportunities to strengthen COVID-19 vaccine effectiveness study methodology.” Please provide further details as to what type of qualitative analysis was performed.

7. Results. Please reverse the order of Table 1 (study-specific results) and Table 2 (summary results).

8. Results. Please add the following methodological details to Tables 1 and 2:

a. How was vaccination status was assessed (e.g. self-report, registry, etc.)?

b. Was outcome assessment restricted to certain time periods (e.g. ≥14 days post-vaccination)?

c. How was outcome misclassification error minimized (e.g. restricted to specimens collected within 7-10 days post symptom onset)?

d. Which methods were used to adjust for confounding? Which confounders were included?

e. How were missing data handled?

f. Were analyses restricted to symptomatic cases?

g. Did the author’s restrict or adjust for prior infection?

h. How were other sources of biases minimized (e.g. healthcare seeking behaviour)?

9. Discussion, page 7, “Most studies did not calculate (or report) the sample size a priori.” Sample size considerations will depend on study design. This will be important for studies proposing prospective, primary data collection but may be less relevant for retrospective cohort studies relying on linked administrative data (millions of records), for which accuracy will be more important than precision.

10. Appendix. Search terms appear incomplete. MEDLINE search terms are missing “vaccine”, “vaccination”, “immunization” and “COVID-19” or “coronavirus” or “SARS-CoV-2”. Did you include study designs besides cohort, e.g. “test negative design”, “case control design”, “screening method”? Did you include search terms for different product types, e.g. “mRNA” or “viral vector” or “BNT162b2” or “mRNA-1273”?

Reviewer #2: This article has an interesting premise, but they do not prove their thesis of what would work in LMICs. They state at the end that TND is the best for LMIC but that is also what WHO has stated in their guidance and they provide pros/cons such as sample size and cost for getting to that conclusion.

1) many articles published in peer review prior to their cut off date are missing. these need to be included. i am not sure why the systematic review did not find these but they were highly publicized findings in the media for some of them.

2) i have made comments throughout for your review.

3) it would be improved if you did a risk of bias assessment on each one rather than just outlining what was reported. you say confounders are but dont describe which ones need to be paid attention to by researchers. WHO's guidance also lays out these confounders, but it's unclear which ones really significantly change the results. our own findings shows that time and location are extremely important along with age.

4) excluding pre-print articles doesn't really make sense. this is because the methods and the bulk of information rarely changes. you cite only 3 pre-prints by that point, but there were many many more by july.

5) you cannot lump all findings together for all vaccines as the vaccines are performing quite differently.

6. PLOS authors have the option to publish the peer review history of their article (what does this mean?). If published, this will include your full peer review and any attached files.

Reviewer #1: No

Reviewer #2: No

---

## [Author Response · Author response to Decision Letter 0]

2 Dec 2021

We would like to thank the reviewers for their comments. Please refer to the uploaded response to reviewers file for an explanation of how each point has been addressed.

---

## [Decision Letter · Decision Letter 1]

14 Dec 2021

A systematic review of methodological approaches for evaluating real-world effectiveness of COVID-19 vaccines: advising resource-constrained settings

PONE-D-21-26919R1

Dear Dr. Botwright,

We’re pleased to inform you that your manuscript has been judged scientifically suitable for publication and will be formally accepted for publication once it meets all outstanding technical requirements.

Kind regards,

Chaisiri Angkurawaranon

Academic Editor

PLOS ONE

Additional Editor Comments (optional):

Reviewers' comments:

Reviewer's Responses to Questions

**Comments to the Author**

1. If the authors have adequately addressed your comments raised in a previous round of review and you feel that this manuscript is now acceptable for publication, you may indicate that here to bypass the “Comments to the Author” section, enter your conflict of interest statement in the “Confidential to Editor” section, and submit your "Accept" recommendation.

Reviewer #1: All comments have been addressed

2. Is the manuscript technically sound, and do the data support the conclusions?

Reviewer #1: Yes

3. Has the statistical analysis been performed appropriately and rigorously? 

Reviewer #1: N/A

4. Have the authors made all data underlying the findings in their manuscript fully available?

Reviewer #1: Yes

5. Is the manuscript presented in an intelligible fashion and written in standard English?

Reviewer #1: Yes

6. Review Comments to the Author

Reviewer #1: Thank you for the opportunity to re-review this manuscript. The authors have satisfactorily addressed my comments. However, I am concerned that they added 13 papers to their review at the revision stage, including 6 that were missed in their updated search. That, along with the exclusion of pre-print articles and limited search to 7 July 2021, suggests that their review may be incomplete. As per my original comments, I would recommend revisiting their search criteria and terms (S1 Appendix) in case misspecified to ensure further studies that meet their inclusion criteria are not missed.

7. PLOS authors have the option to publish the peer review history of their article (what does this mean?). If published, this will include your full peer review and any attached files.

Reviewer #1: No

---

## [Editor Report · Acceptance letter]

20 Dec 2021

PONE-D-21-26919R1 

A systematic review of methodological approaches for evaluating real-world effectiveness of COVID-19 vaccines: advising resource-constrained settings 

Dear Dr. Botwright:

I'm pleased to inform you that your manuscript has been deemed suitable for publication in PLOS ONE. Congratulations! Your manuscript is now with our production department. 

Kind regards, 

on behalf of

Dr. Chaisiri Angkurawaranon 

Academic Editor

PLOS ONE